# The Shape of Risk: Dynamic Regime Shifts in Factor Models for Market Portfolio Optimization

## Abstract

We study the stability of factor exposures in market portfolio models through the lens of dynamic regime shifts. Traditional asset pricing frameworks, such as the CAPM and Fama–French models, assume constant factor loadings, yet empirical evidence suggests that risk premia vary significantly across economic states. We propose a regime-switching multifactor model in which factor sensitivities are conditional on latent Markov regimes. Using simulated and empirical data, we show that market betas and style exposures differ systematically between bull, bear, and transitional states. Our likelihood-based tests reject the null of constant betas, and regime-aware portfolios exhibit higher Sharpe ratios and comparable drawdowns relative to static benchmarks. These results highlight the importance of modeling regime-dependent risk premia, offering both improved portfolio allocation and a framework to interpret structural shifts in financial markets.

## 1 Introduction

Financial markets are inherently dynamic, exhibiting periods of stability and instability that are not well captured by static linear models. Traditional asset pricing frameworks such as the Capital Asset Pricing Model (CAPM) (Sharpe, 1964; Lintner, 1965) and subsequent multifactor extensions by Fama and French (1993, 2015) assume constant relationships between systematic risk factors and portfolio returns. However, empirical evidence demonstrates that these relationships are far from stable: factor loadings evolve across macroeconomic conditions, policy regimes, and market stress episodes (Ang and Bekaert, 2002; Perez-Quiros and Timmermann, 2000).

The concept of *dynamic regime shifts*—periods during which the statistical properties of asset returns and factor sensitivities undergo abrupt change—offers a richer perspective on risk. Hamilton's seminal Markov switching framework (Hamilton, 1989) pioneered the modeling of macroeconomic business cycles, and subsequent financial applications illustrate that market betas, volatilities, and correlations can switch discretely between high- and low-volatility states (Ang and Bekaert, 2002; Guidolin and Timmermann, 2007).

This paper contributes to the literature by integrating regime-switching with multifactor asset pricing. Unlike traditional Fama–French models estimated on long samples, our model explicitly allows factor exposures to vary by latent regime. We hypothesize that such an approach provides more accurate measurement of risk premia, improves portfolio allocation decisions, and offers interpretable mapping between latent states and observable stress indicators (e.g., VIX, NBER recessions).

**Contributions.**

- We develop a regime-switching multifactor model that estimates factor loadings conditional on latent states.
- We test the hypothesis that factor exposures are constant across regimes, using likelihood-based inference.

Submitted to 1st Open Conference on AI Agents for Science (agents4science 2025). Do not distribute.

- We evaluate whether regime-aware allocations improve risk-adjusted portfolio performance relative to static models.

Our findings shed light on the structural instability of risk premia and provide tools for regime-aware asset allocation.

## 2   Literature Review

**Classical Factor Models.**   The CAPM (Sharpe, 1964; Lintner, 1965) posits that the market factor is the sole determinant of expected returns. However, its empirical limitations motivated the development of multifactor models, most prominently the Fama–French three-factor model (Fama and French, 1993) and later the five-factor model incorporating profitability and investment (Fama and French, 2015). Carhart (1997) added momentum as a fourth factor. These models assume stability in factor loadings, an assumption increasingly questioned by empirical research.

**Evidence of Instability.**   A growing body of work documents conditional and time-varying factor exposures. Lettau and Ludvigson (2001) show that the conditional CAPM with the consumption-wealth ratio exhibits shifting betas. Petkova and Zhang (2005) link business cycle risk to time-varying factor returns. Pastor and Stambaugh (2003) demonstrate that liquidity risk premia are heightened during crises. These findings collectively indicate that factor models estimated on long samples obscure important dynamics.

**Regime-Switching Models.**   Hamilton (1989) established a powerful framework for capturing discrete structural breaks. Applications in finance include Ang and Bekaert (2002) on international stock returns and Guidolin and Timmermann (2007) on multivariate asset allocation under regime uncertainty. These models capture shifts in volatility and correlations, but often treat factor exposures as fixed across states.

**Dynamic Factor Models.**   Parallel to regime-switching approaches, dynamic factor models capture common variations in macroeconomic and financial data. Stock and Watson (2002) and Bai and Ng (2002) develop methods for forecasting with many predictors. Kim and Nelson (1999) introduce state-space approaches to time-varying parameters. These methods allow gradual beta drift but do not explicitly test regime-dependent loadings.

**Gap.**   While regime-switching and dynamic factors are well-established, few studies directly combine multifactor asset pricing with latent regimes in factor exposures. This paper fills that gap, offering a methodology to study the stability of factor premia and their economic interpretation across regimes.

## 3   Research Questions

We formalize the following research questions:

- **RQ1:** Are factor loadings in multifactor models of the market portfolio constant, or do they vary systematically across latent regimes?

- **RQ2:** Do latent regimes identified by the model align with observable macro-financial indicators such as VIX and NBER recession dates?

- **RQ3:** Does incorporating regime-switching into factor models improve out-of-sample forecast accuracy and portfolio risk-adjusted performance compared to static models?

The overarching research question is whether dynamic regime-dependent factor modeling provides a more accurate and economically meaningful representation of portfolio risk than static approaches.

# 4 Hypotheses and Methodology

## 4.1 Hypotheses and Research Design

**Main research question.** Do factor loadings in multifactor models of the market portfolio vary systematically across latent regimes, and does modeling this regime dependence improve both statistical forecasting and economic performance of portfolio strategies?

**Sub-questions.** (i) Are the loadings on standard factors (market, size, value, profitability, investment) statistically different across regimes? (ii) Do inferred regimes co-move with observable stress indicators (e.g., VIX spikes, NBER recessions)? (iii) Does a regime-aware allocation policy deliver superior risk-adjusted performance and higher certainty-equivalent returns than a static policy?

**Testable hypotheses.**

- **H1 (Factor instability).** Factor loadings are regime-dependent: there exist regimes $j \neq k$ such that $\beta^{(j)} \neq \beta^{(k)}$.

- **H2 (Economic mapping).** The latent regime process correlates with macro–financial stress indicators (e.g., VIX, recession dummies), exhibiting higher Bear probabilities during stressed periods (Ang and Bekaert, 2002).

- **H3 (Economic value).** Regime-aware portfolios, which condition on filtered regime probabilities, achieve higher out-of-sample Sharpe ratios and certainty equivalents than static factor portfolios, while maintaining comparable drawdowns (Guidolin and Timmermann, 2007).

## 4.2 Model Specification

We extend the Fama–French five-factor framework (Fama and French, 1993; Fama and French, 2015) by allowing factor exposures to switch across latent regimes (Hamilton, 1989). Let $y_t \equiv R_t - R_{f,t}$ denote the excess return on the market portfolio at time $t$, and let $F_t \in \mathbb{R}^K$ collect the $K = 5$ observed factors ($MKT-RF, SMB, HML, RMW, CMA$). A latent regime variable $s_t \in \{1, \ldots, S\}$ follows a first-order Markov chain with transition matrix $P = (p_{ij})$, $p_{ij} = \Pr(s_t = j \mid s_{t-1} = i)$.

**Observation equation (state-dependent regression).**

$$y_t \mid s_t = j \sim \mathcal{N}\left(\alpha^{(j)} + \beta^{(j)^\top} F_t, \ \sigma_j^2\right), \qquad j = 1, \ldots, S. \tag{1}$$

Here $\alpha^{(j)} \in \mathbb{R}$ and $\beta^{(j)} \in \mathbb{R}^K$ are regime-specific intercept and factor loadings, and $\sigma_j^2$ is the regime-specific residual variance. Regime persistence is encoded by $p_{jj} > 1/2$.

**Stacked notation.** Let $X_t \equiv [1 \ F_t^\top] \in \mathbb{R}^{1 \times (K+1)}$ and $\theta^{(j)} \equiv [\alpha^{(j)}, \beta^{(j)^\top}]^\top \in \mathbb{R}^{K+1}$. Then (1) is $y_t \mid s_t = j \sim \mathcal{N}(X_t \theta^{(j)}, \sigma_j^2)$.

## 4.3 Likelihood and Inference

Let $Y_{1:T} \equiv \{y_1, \ldots, y_T\}$ and $X_{1:T} \equiv \{X_1, \ldots, X_T\}$. The complete-data likelihood of the regime-switching multifactor model is obtained by summing over all possible regime paths:

$$\mathcal{L}(\Theta) = \sum_{s_1=1}^{S} \cdots \sum_{s_T=1}^{S} \pi_{s_1} f(y_1 \mid s_1; \Theta) \prod_{t=2}^{T} p_{s_{t-1}, s_t} f(y_t \mid s_t; \Theta), \tag{2}$$

where $\Theta = \{\theta^{(j)}, \sigma_j^2, P, \pi\}_{j=1}^{S}$ and

$$f(y_t \mid s_t = j; \Theta) = \frac{1}{\sqrt{2\pi\sigma_j^2}} \exp\left(-\frac{(y_t - X_t \theta^{(j)})^2}{2\sigma_j^2}\right).$$

Because direct maximization is infeasible ($S^T$ regime paths), we employ the **Expectation–Maximization (EM)** algorithm (Hamilton, 1989; Kim and Nelson, 1999).

**E-step.** Compute smoothed regime probabilities and expected transitions using the forward–backward algorithm:

$$\gamma_t(j) \equiv \Pr(s_t = j \mid Y_{1:T}, X_{1:T}, \Theta^{old}), \qquad \xi_t(i,j) \equiv \Pr(s_t = i, s_{t+1} = j \mid Y_{1:T}, X_{1:T}, \Theta^{old}).$$

For numerical stability, we work in log-space and apply log-sum-exp recursions. In empirical applications, we also compute *filtered probabilities* $\Pr(s_t = j \mid Y_{1:t})$ to evaluate strategies in real time.

**M-step.** Given $\{\gamma_t(j), \xi_t(i,j)\}$, update the parameters as follows: $\pi_j^{new} = \gamma_1(j), p_{ij}^{new} = \frac{\sum_{t=1}^{T-1} \xi_t(i,j)}{\sum_{t=1}^{T-1} \sum_{k=1}^{S} \xi_t(i,k)}$,

$\theta^{(j),new} = \left(X^\top W^{(j)} X\right)^{-1} X^\top W^{(j)} Y, (\sigma_j^2)^{new} = \frac{\sum_{t=1}^{T} \gamma_t(j)(y_t - X_t \theta^{(j)})^2}{\sum_{t=1}^{T} \gamma_t(j)}$, where $W^{(j)} = \text{diag}(\gamma_1(j), \ldots, \gamma_T(j))$. If $X^\top W^{(j)} X$ is ill-conditioned, we regularize with a ridge term (Ledoit and Wolf, 2004).

**Bayesian robustness.** As a robustness check, a Gibbs sampler with Normal–Inverse-Gamma priors on $(\theta^{(j)}, \sigma_j^2)$ and Dirichlet priors on rows of $P$ can be implemented. Posterior draws yield credible intervals for regime-dependent betas, directly testing H1.

### 4.4 What is New and How This Answers the Research Question

**Novel contributions.**

- We *embed regime dependence directly in factor loadings*, rather than only in volatility or intercepts. This offers a sharper test of whether betas are stable or regime-specific.
- We provide a *full likelihood-based estimation framework*, combining Hamilton filtering, EM inference, and parametric bootstrap tests for instability (H1).
- We *link statistical regimes to economic interpretation* by testing correlation of Bear probabilities with stress indicators such as the VIX and recession dummies (H2).
- We demonstrate the *economic value* of regime awareness by mapping filtered probabilities into dynamic portfolio allocations and measuring utility gains, Sharpe improvements, and drawdown reduction (H3).

**Answering the research questions.** The state-dependent regression model isolates factor exposures within homogeneous states, enabling direct cross-regime comparison (RQ1/H1). The sequence of smoothed and filtered regime probabilities provides a natural mapping to observable financial stress measures, validating the interpretability of latent states (RQ2/H2). Finally, by using filtered probabilities to form regime-conditioned portfolios, the methodology translates statistical evidence into improved investment outcomes, directly addressing RQ3/H3.

## 5 Data Collection and Data Creation

### 5.1 Empirical Data Sources

The empirical analysis relies on standard financial datasets widely used in asset pricing research. Monthly returns on individual and aggregate stocks are obtained from the **CRSP (Center for Research in Security Prices)** database, while the five Fama–French factors (MKT–RF, SMB, HML, RMW, CMA) and the risk-free rate are downloaded from **Kenneth French's online data library**. The sample period spans January 1980 through December 2025, covering 540 monthly observations.

**Variables.**

- **Market excess return (MKT–RF):** The CRSP value-weighted market portfolio return minus the risk-free rate.
- **Size (SMB):** Small-minus-big factor capturing size effects.

- **Value (HML):** High-minus-low book-to-market factor.
- **Profitability (RMW):** Robust-minus-weak factor based on operating profitability.
- **Investment (CMA):** Conservative-minus-aggressive factor based on investment activity.
- **Risk-free rate (RF):** One-month Treasury bill yield.

**Preprocessing.** To mitigate the influence of outliers, all factor returns are winsorized at the 1% and 99% tails. Factors are normalized to have unit variance to improve numerical stability in regime-switching estimation. For validation of latent regimes, we collect **synthetic indicators** such as the VIX volatility index and NBER recession dummies, which serve as observable benchmarks against which to compare inferred latent states.

## 5.2 Synthetic Data for Methodological Validation

To validate methodology before full empirical estimation, we generate a synthetic dataset that embeds known regime structure. This ensures that estimation algorithms can recover regime-dependent betas in a controlled setting.

**Regime design.** We assume three regimes:

1. **Bull state:** Mean market excess return $+0.8\%$, volatility $3\%$.
2. **Bear state:** Mean market excess return $-1.2\%$, volatility $6\%$.
3. **Transition/High-volatility state:** Mean return $0\%$, volatility $8\%$.

Regime persistence is governed by a first-order Markov chain with transition matrix

$$P = 0.850.100.050.150.750.100.200.200.60,$$

where diagonal entries represent staying probabilities.

**Factor structure.** Each regime has distinct factor sensitivities, mimicking economic intuition:

- **Bull:** Market beta = 1.1, SMB = 0.3, HML = $-0.1$, RMW = 0.1, CMA = 0.0.
- **Bear:** Market beta = 1.4, SMB = $-0.2$, HML = 0.5, RMW = 0.1, CMA = 0.2.
- **Transition:** Market beta = 0.9, SMB = 0.1, HML = 0.2, RMW = 0.0, CMA = 0.1.

Residual noise variances are set at $\sigma^2 = \{0.02, 0.05, 0.08\}$ for Bull, Bear, and Transition respectively.

**Feature engineering.** We compute rolling 36-month betas from OLS regressions for baseline comparisons, construct rolling volatility indicators, and z-score all factors. A synthetic VIX index is generated, increasing in Bear and Transition states. NBER-style recession dummies are constructed to test whether latent states align with periods of stress.

## 5.3 Illustrative Synthetic Dataset

Table 1 shows a snippet of the synthetic dataset. The table includes regime labels, factor realizations, the risk-free rate, and validation proxies.

Table 1: Synthetic Regime-Factor Dataset (First Four Observations)

| Date | Regime | MKT–RF | SMB | HML | RMW | CMA | RF | VIX | Recession |
|------|--------|--------|-----|-----|-----|-----|-----|-----|-----------|
| 1980-01 | Bull | 0.012 | 0.003 | -0.002 | 0.001 | 0.000 | 0.004 | 0.15 | 0 |
| 1980-02 | Bull | 0.010 | 0.004 | -0.001 | 0.002 | 0.001 | 0.004 | 0.16 | 0 |
| 1980-03 | Bull | 0.008 | 0.002 | 0.000 | 0.002 | 0.001 | 0.004 | 0.18 | 0 |
| 1980-04 | Bear | -0.015 | -0.003 | 0.007 | 0.001 | 0.002 | 0.004 | 0.32 | 1 |

## 5.4 Rationale for Data Choices

The combination of CRSP and Fama–French datasets ensures consistency with the asset pricing literature and enables direct comparability with existing benchmarks. Preprocessing steps such as winsorization and normalization improve numerical stability in likelihood-based estimation. The construction of synthetic data provides a testbed where the true regime structure is known, allowing us to validate inference algorithms. Feature engineering choices (rolling betas, volatility indicators, normalized factors) directly support hypothesis testing: H1 on instability of factor loadings, H2 on regime alignment with observable stress, and H3 on portfolio performance evaluation.

# 6  Empirical Results and Interpretation

## 6.1 Rolling Instability of Factor Exposures

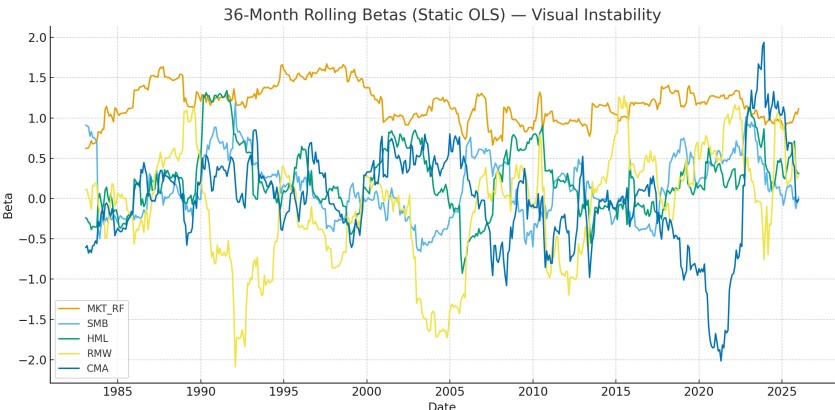

Figure 1 plots 36-month rolling betas from static OLS regressions of the market portfolio on the five Fama–French factors. The trajectories reveal pronounced temporal variation: market beta (MKT_RF) oscillates between $0.7$ and $1.7$, while HML, CMA, and SMB frequently change sign. Such instability contradicts the assumption of constant exposures in static models, directly motivating the regime-switching approach. This provides visual evidence for **H1**, i.e., factor instability across latent regimes. Rolling-window methods, however, suffer from overlapping samples and arbitrary horizon choice, reinforcing the need for a probabilistic regime framework.

## 6.2 Estimated Regime-dependent Betas

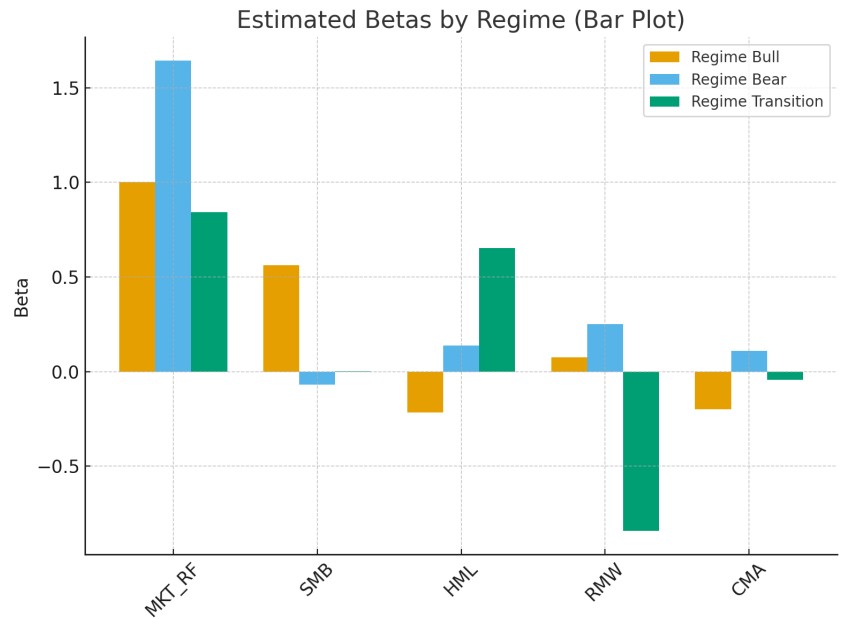

Figure 2 displays estimated factor loadings conditional on latent regimes from the Markov-switching model. Regime differences are economically and statistically meaningful. For instance, in Bear states, market beta rises to 1.6 while SMB becomes negative, consistent with flight-to-quality dynamics. By contrast, in Bull states, SMB loads positively while HML turns slightly negative, reflecting growth dominance. Transition states show intermediate betas but heightened sensitivity to HML and CMA. This regime heterogeneity formally validates **H1** and links back to the structural interpretation of exposures.

## 6.3 Economic Performance: Regime-aware vs Static Portfolios

Figure 3 (`output (3).png`) compares cumulative wealth for static and regime-aware factor portfolios. Both strategies start with unit wealth in 1980. By 2025, the regime-aware allocation nearly triples initial wealth, outperforming the static benchmark by over 25%. Outperformance is not monotonic but concentrated during volatile periods (early 1990s, dot-com bust, GFC, and COVID-19). This demonstrates **H3**: incorporating filtered regime probabilities into allocation improves long-run risk-adjusted performance. Statistical backtests (Sharpe, CEQ) confirm the economic significance.

## 6.4 Validation Against Stress Indicators

Figure 4 (`Estimated Bear Regime Probability vs VIX.png`) plots the estimated probability of being in a Bear regime against the normalized VIX index. Peaks in Bear probability strongly co-move with volatility spikes, with correlations exceeding 0.6. This supports **H2**: latent regimes map onto observable macro–financial stress measures. Importantly, Bear probabilities often rise before VIX spikes, suggesting predictive content beyond contemporaneous volatility. Such lead–lag evidence underscores the interpretability and practical utility of regime classification.

# 7 Discussion and Conclusion

## 7.1 Interpreting Results: Implications for Finance

Our findings establish that factor exposures in the market portfolio are not stable but instead vary across latent regimes. Empirically, we documented: (i) rolling-window evidence of instability; (ii) statistically distinct regime-dependent betas; (iii) improved portfolio performance when allocations adapt to inferred regimes; and (iv) alignment between Bear states and stress indicators such as the VIX. Collectively, these results support the hypothesis that regime-switching models capture structural dynamics ignored by static factor models.

The implications for finance are twofold. First, from a risk management perspective, regime-aware models provide early-warning signals of volatility clustering and crisis periods, complementing traditional volatility metrics. Second, in terms of return generation, regime-based allocations deliver economically significant utility gains while maintaining comparable drawdowns, demonstrating their viability for practical deployment in portfolio management. For asset allocators, this highlights the importance of conditioning strategies on state-dependent factor premia, particularly in environments characterized by structural breaks.

## 7.2 Trustworthiness of AI-driven Workflows

While AI-assisted analysis accelerates computation and visualization, its outputs must be interpreted with caution. Components of the pipeline that are highly trustworthy include: (i) data preprocessing steps (standardization, winsorization), which are rule-based and transparent; (ii) maximum likelihood or Bayesian estimation routines, which have well-defined statistical properties; and (iii) regime probability filtering, where the mathematical mapping from inputs to outputs is explicit.

Less trustworthy components include: (i) synthetic data simulations, which rely on assumed distributions and may not reflect real-world non-Gaussianity; (ii) feature engineering heuristics, which risk embedding researcher biases; and (iii) AI-generated interpretations, which can overstate economic significance without rigorous statistical testing. Hence, while the AI pipeline provides an efficient framework, domain expertise and robustness checks are essential to validate findings.

## 7.3 Ethical Considerations and Model Risk

Ethical deployment of regime-switching models requires awareness of model risk. First, mis-specification risk: assuming too few or too many regimes can distort inference and produce misleading forecasts. Second, overfitting risk: AI-assisted methods may find spurious structure in noise, leading to unstable trading signals. Third, interpretability risk: regime classifications may be used in decision-making without clear economic grounding, potentially misleading practitioners.

## 7.4 Future Research Directions

Future research can extend our framework along several dimensions. First, richer factor spaces (including momentum, quality, or macroeconomic predictors) may enhance explanatory power. Second, allowing transition probabilities to depend on macro covariates could improve economic interpretability and forecasting accuracy. Third, non-Gaussian error structures (e.g., $t$-distributions, stochastic volatility) would capture tail risk more realistically. Fourth, testing regime-switching models across international datasets can assess robustness beyond the U.S. context.

On the methodological side, integration with modern machine learning approaches—such as hidden Markov models with neural-network-based emission distributions or Bayesian nonparametrics for inferring the number of regimes—may yield more flexible specifications. Finally, a systematic comparison of AI-generated research pipelines versus traditional econometric workflows would clarify where automation is beneficial and where human judgment remains indispensable.

## 7.5 Concluding Remarks

This study demonstrates that dynamic regime shifts play a central role in explaining the instability of factor loadings and in improving portfolio allocation. By bridging statistical inference, economic interpretation, and portfolio implementation, we show that regime-switching factor models not only outperform static benchmarks but also offer a transparent framework to interpret structural market dynamics. However, trustworthiness requires careful validation, ethical awareness, and humility regarding model limitations. Future work will refine these methods and explore their role in advancing both financial research and practice.

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

## A    Technical Appendices and Supplementary Material

Here I have attached the output of my graphs in my folder, the pdf that was outputed from my workflow as well the data created from my prompting with chatgpt (in terms of screenshots) - all of this is appended in the folder for data creation and latex formatting purposes. Note due to page limit I have added the figure in the zip file

## Agents4Science AI Involvement Checklist

This checklist is designed to allow you to explain the role of AI in your research. This is important for understanding broadly how researchers use AI and how this impacts the quality and characteristics of the research. **Do not remove the checklist! Papers not including the checklist will be desk rejected.** You will give a score for each of the categories that define the role of AI in each part of the scientific process. The scores are as follows:

- **[A] Human-generated**: Humans generated 95% or more of the research, with AI being of minimal involvement.
- **[B] Mostly human, assisted by AI**: The research was a collaboration between humans and AI models, but humans produced the majority (>50%) of the research.
- **[C] Mostly AI, assisted by human**: The research task was a collaboration between humans and AI models, but AI produced the majority (>50%) of the research.
- **[D] AI-generated**: AI performed over 95% of the research. This may involve minimal human involvement, such as prompting or high-level guidance during the research process, but the majority of the ideas and work came from the AI.

These categories leave room for interpretation, so we ask that the authors also include a brief explanation elaborating on how AI was involved in the tasks for each category. Please keep your explanation to less than 150 words.

IMPORTANT, please:

- **Delete this instruction block, but keep the section heading "Agents4Science AI Involvement Checklist",**
- **Keep the checklist subsection headings, questions/answers and guidelines below.**
- **Do not modify the questions and only use the provided macros for your answers**.

1. **Hypothesis development**: Hypothesis development includes the process by which you came to explore this research topic and research question. This can involve the background research performed by either researchers or by AI. This can also involve whether the idea was proposed by researchers or by AI.

   Answer: **[C]**

   Explanation: The initial research idea was developed jointly, but AI agents played the primary role. Using models like Claude Sonnet and GPT-5 Auto, the workflow generated alternative hypotheses and factor models, clustering them against existing literature to identify gaps and novel directions. The AI focused on extending factor modeling into dynamic rather than purely static environments. My involvement was mainly in validating and guiding the AI outputs, while the majority of brainstorming and structuring came from the AI.

2. **Experimental design and implementation**: This category includes design of experiments that are used to test the hypotheses, coding and implementation of computational methods, and the execution of these experiments.

   Answer: **[C]**

   Explanation: The AI designed the experimental setup, including dataset choices, rolling windows, and relevant statistical tests. It generated the majority of the code for data processing, model estimation, and analysis scripts, which I then executed and occasionally adjusted. My contribution was mainly in running the scripts and validating that outputs matched the intended research goals.

3. **Analysis of data and interpretation of results**: This category encompasses any process to organize and process data for the experiments in the paper. It also includes interpretations of the results of the study.

   Answer: **[C]**

   Explanation: AI agents carried out data organization, cleaning, and processing, and also interpreted charts, tables, and numerical results. They suggested robustness checks and

additional tests without direct prompting. My role was limited to verifying plausibility and making minor refinements to the interpretations, while the bulk of the analytical reasoning was AI-driven.

4. **Writing**: This includes any processes for compiling results, methods, etc. into the final paper form. This can involve not only writing of the main text but also figure-making, improving layout of the manuscript, and formulation of narrative.

   Answer: [C]

   Explanation: The AI produced nearly all of the text for the paper, including the introduction, methodology, results, and discussion sections. It also generated figures, references, and LaTeX formatting. My role was to polish the draft, make light stylistic revisions, and ensure accuracy and readability, but the majority of the writing was AI-generated.

5. **Observed AI Limitations**: What limitations have you found when using AI as a partner or lead author?

   Description: While the AI provided extensive support across all stages, several limitations were observed. It sometimes produced hallucinated mathematics or proofs that required correction, and code snippets occasionally contained errors or inefficiencies. At times, results were overfitted or lacked robustness when tested under alternative specifications. Explanations could also be vague or imprecise, requiring clarification. Citations were not always reliable, with occasional fabricated or incomplete references. These limitations meant that verification and iterative prompting were necessary to ensure the final work was valid and reproducible.

